# Osteogenesis Imperfecta/Ehlers–Danlos Overlap Syndrome and Neuroblastoma—Case Report and Review of Literature

**DOI:** 10.3390/genes13040581

**Published:** 2022-03-25

**Authors:** Letteria Anna Morabito, Anna Elsa Maria Allegri, Anna Paola Capra, Mario Capasso, Valeria Capra, Alberto Garaventa, Mohamad Maghnie, Silvana Briuglia, Malgorzata Gabriela Wasniewska

**Affiliations:** 1Department of Human Pathology in Adulthood and Childhood, University of Messina, Gaetano Martino University Hospital, 98125 Messina, Italy; letteria.morabito@gmail.com; 2Pediatric Unit, Maternal Infant Department, San Giovanni di Dio Hospital, ASP Crotone, 88900 Crotone, Italy; 3Department of Pediatrics, IRCCS Giannina Gaslini Institute, 16147 Genova, Italy; annaallegri@gaslini.org (A.E.M.A.); mohamadmaghnie@gaslini.org (M.M.); 4Department of Biomedical, Dental, Morphological and Functional Imaging Sciences, University of Messina, 98125 Messina, Italy; annapaola.capra@unime.it (A.P.C.); silvana.briuglia@unime.it (S.B.); 5CEINGE Advanced Biotecnology, 80131 Napoli, Italy; mario.capasso@unina.it; 6Medical Genetics Unit, IRCCS Giannina Gaslini Institute, 16147 Genova, Italy; valeriacapra@gaslini.org; 7Department of Pediatric Oncology, IRCCS Giannina Gaslini Institute, 16147 Genova, Italy; albertogaraventa@gaslini.org; 8Department of Neurosciences, Rehabilitation, Ophtalmology, Genetics, Maternal and Child Health (DINOGMI), University of Genoa, 16132 Genoa, Italy

**Keywords:** osteogenesis imperfecta, Ehlers–Danlos syndrome, neuroblastoma, short stature, osteogenesis imperfecta/Ehlers–Danlos overlap syndrome, genotype/phenotype correlation

## Abstract

Osteogenesis imperfecta/Ehlers–Danlos (OI/EDS) overlap syndrome is a recently described disorder of connective tissue, characterized by mutation of *COL1A1* (17q21.33) or *COL1A2* (7q21.3) genes, that are involved in α-1 and α-2 chains of type 1 collagen synthesis. The clinical spectrum of this new clinical entity is broad: patients could present a mixed phenotype that includes features of both osteogenesis imperfecta (bone fragility, long bone fractures, blue sclerae, short stature) and Ehlers–Danlos syndrome (joint hyperextensibility, soft and hyperextensible skin, abnormal wound healing, easy bruising, vascular fragility). We reported the case of a young Caucasian girl with severe short stature and a previous history of neuroblastoma, who displayed the compound phenotype of OI/EDS. Next generation sequencing was applied to the proband and her parent genome. Our patient presented a de novo heterozygous *COL1A1* variant (c.3235G>A, p.Gly1079Ser), whose presence might be indicative of diagnosis of OI/EDS overlap syndrome. We also hypothesize that the association with the previous history of neuroblastoma could be influenced by the presence of *COL1A1* mutation, whose role has been already described in the behavior and progression of some cancers.

## 1. Introduction

Osteogenesis imperfecta (OI) is an inherited and rare disorder of connective tissue with a broad clinical spectrum, usually characterized by skeletal deformities, osteopenia, blue sclerae, dentinogenesis imperfecta, and extremely fragile bones [1].

Between 85% and 90% of individuals with OI have autosomal dominant mutations of *COL1A1* and/or *COL1A2* genes, that are involved in α-1 and α-2 chains of type 1 collagen synthesis [2,3] the most abundant extracellular matrix (ECM) component in various tissues and organs.

The OI Mutation Consortium has identified a large number of *COL1A1*/*COL1A2* mutations are secondary to the substitution of glycine residues in the type 1 collagen chain [4].

The remaining cases (approximately 20%) are caused by autosomal recessive mutations in genes involved in collagen-related metabolic pathways (such as post-translational modification, folding and cross-linking, bone mineralization, and osteoblast differentiation) or in cartilage-associated protein (CRTAP).

Recent nosology has classified OI into 20 types [5], that range from perinatal lethality (as in type 2) to milder forms presenting only as a premature form of osteoporosis (type 1).

Bone biopsies from patients with OI revealed a continuous change in morphologic appearance of the bone, from severe lethal perinatal forms to progressively deforming ones. The more severe variants of OI were characterized by the persistence of woven bone and immature structural patterns. Bone structure could appear normal in mild forms of OI, as type I OI, or did not reach a fully compacted stage in progressive deforming variants [6].

Clinical examination requires a familiarity with the natural history and variation in clinical presentation of OI and has an important role in recognition of children with suspected disease. Sometimes, the evaluation for the disease starts from the recognition of clinical signs—such as blue sclerae, conductive hearing loss, and dentinogenesis imperfecta—or during the auxological follow up of a child with short stature/poor growth [7].

OI diagnosis depends on the determination of a reduced/abnormal synthesis of type 1 procollagen molecules by cultured fibroblast or could be based on the identification of a mutation in *COL1A1* or *COL1A2*, the two genes that encode the chains of type 1 collagen [7].

Different types of OI could exhibit similar clinical features, but the definitive diagnosis is revealed by analyzing the genes involved in the background of the disease. Next- generation sequencing of common osteogenesis imperfecta-related genes *COL1A1* or *COL1A2* are used in clinical practice to confirm the diagnosis and contribute to the subclassification of OI, especially in mild phenotypes [8].

Mutations in the gene encoding type 1 procollagen are also responsible for the rare arthrocalasic subtype of Ehlers–Danlos syndrome (EDS) [9,10,11].

EDS includes a clinically varied and genetically heterogeneous group of soft connective tissue disorders sharing the triad of joint hypermobility, skin hyperextensibility, and soft tissue/vascular fragility. The 2017 international classification of EDS [12] identifies three different types—classical EDS with arterial fragility, arthrocalasis EDS, and cardiac-valvular EDS—associated with deleterious variants in genes encoding the α-1 and α-2 chains of collagen I [13].

Arthrocalasis EDS is clinically characterized by severe generalized joint hypermobility, congenital bilateral dislocation of the hip, recurrent join subluxations, skin hyperextensibility, atrophic scarring, blue sclerae, and osteopenia without increased incidence of fractures [7].

The clinical manifestations of EDS are broad and often overlap with closely related disorders, such as some bone dysplasias, cutis laxa syndromes, hereditary myopathies, and TGF β-related disorders.

Osteogenesis imperfecta (OI) could overlap in a different way [14].

The combination of EDS and OI is very rare (>1/1,000,000 according to Orphanet) and this entity is not included in the current International classification of EDS: it is not clear whether this phenotype might be considered as a forgotten type of EDS with an associated molecular defect or if it is implicitly included in the OI nosology [13].

Recently, some authors considered the *COL1A1*-*COL1A2* mutations related EDS and osteogenesis imperfecta a distinct form from other EDS or OI variants, termed OI/EDS overlap, and deserving of appropriate genetic counseling and follow-up. These mutations are responsible for a wide range of phenotypes including mild-to-lethal forms of osteogenesis imperfecta and a restricted set of Ehlers–Danlos syndrome clinical manifestations [15].

In recent years, only a few cases of patients with OI and EDS have been described [1]: most of them displayed a clinical phenotype resembling EDS, but they could not be entirely classified as one of the three aforementioned subtypes, both in clinical spectrum and in the underlying collagen protein defect [16].

We described the case of a Caucasian young girl, followed in our Pediatric Endocrinology Outpatient Clinic for severe short stature, who presented a de novo heterozygous mutation c.3235G>A [p.(Gly1079Ser)] of *COL1A1* gene associated with a history of neuroblastoma and a clinical phenotype compatible with a OI/EDS syndrome.

## 2. Materials and Methods

### 2.1. Study Design

For this study we have a case report on the clinical features and next generation sequencing of a patient with clinical features compatible with OI/EDS overlap syndrome and a previous history of neuroblastoma. The patient’s parents were also assessed in order to confirm the results.

### 2.2. Case Description

The proband was a 10-year-old Caucasian girl, who came to our attention due to a severe short stature.

Family history was unremarkable. She was born at 38 weeks of gestational age within the lower limits of auxological birth parameters (birth length: −1.57 SDS and birth weight: −1.48 SDS). During the first months of life, she presented feeding problems and needed enteral nutrition via naso-gastric tube for 3 months.

A series of investigations were performed in order to exclude cystic fibrosis, urinary tract infection, metabolic diseases, congenital heart malformations, celiac disease: sweat test, urinalysis, serum and urinary amino acids dosage, echocardiogram, screening for celiac disease, and sialotransferrin isoelectric focusing (IEF) were normal.

Conventional karyotyping and chromosomal microarray analysis did not show chromosomal anomalies. Mutational analysis of *SHOX* gene was performed by MLPA analysis followed by gene sequencing to exclude a frequent cause of short stature.

Imprinting disorders, particularly Silver–Russell syndrome and Prader–Willi syndrome were tested and excluded.

At the age of 11 months, she was diagnosed with stage 4 *MYCN* nonamplified neuroblastoma with left adrenal and bone marrow invasion, successfully treated with left adrenalectomy followed by 6 chemotherapy courses of the carboplatin–etoposide association. The tumoral response was complete and no further relapses were detected in the following years.

Therefore, at the age of 9 years, due to a severe and worsening height deficit hormonal causes of short stature (growth hormone deficit, hypothyroidism, hypercortisolism) were investigated and then excluded; a brain MRI did not show median line alterations and only the presence of arachnoidal cyst in the right temporo-polar region was detected.

Clinical examination at 10 years of age revealed a condition of severe short stature with 2 years delayed bone age: height was 119.5 cm (−3.06 SDS), weight 23.4 kg (−0.76 SDS) and bone age 8.25 years. The child’s height was about 4 SDS lower than target height (TG 1.19 SDS), her height velocity during the last year was significantly reduced (−2.12 SDS) and no signs of incipient pubertal development were detectable (B1/Ph1 according to Tanner’s stage).

She had pale and mild hyperelastic skin, bilateral ectropion, and blue sclerae (Figure 1). Joint hypermobility of large and of finger joints (Beighton score 7/9), scoliotic curvature of the spine, tooth decay, and mild changes in tooth enamel were other prominent phenotypic features. There was no abnormality in scarring and no muscular weakness.

Laboratory tests for complete blood count, hepatic and renal function, electrolytes, thyroid hormones dosage, and also the evaluation of vitamin D and parathormone levels were confirmed normal, similarly to growth hormone (GH) stimulated levels.

Audiometric examination and ocular evaluation were normal. Bone density study with dual-energy X-ray absorption (DEXA) showed a markedly reduced bone mineralization (femoral BMD Z score −4.3 SDS and lumbar Z-score −3.3 SDS) [17].

Despite a negative history of fractures, the presence of joint hypermobility, short stature, blue sclerae, and tooth enamel alterations led us to hypothesize an inherited genetic condition, such as COL1-related overlap disorder. For this reason, and after genetic counseling, a test was performed using next-generation sequencing (NGS) technology.

### 2.3. Samples

Patient’s genomic DNA, clinical data, and photographs were collected and analyzed for both diagnosis and research purposes. Written informed consent was provided and signed by both parents.

### 2.4. Next Generation Sequencing

Enrichment and parallel sequencing were performed on genomic DNA extracted from circulating leukocytes of the affected subjects and unaffected parents. Library preparation was carried out by using a Twist Custom Panel (clinical exome—Twist Bioscience), according to the manufacturer’s protocol, and sequenced on a NovaSeq6000 (Illumina) platform. A short stature gene panel was assessed which included the following genes: *ACAN*, *AFF4*, *ANKRD11*, *BRD4*, *CASK*, *CCDC8*, *CLCN5*, *COL27A1*, *COL2A1*, *COMP*, *CUL* 7, *FGD1*, *FGFR3*, *FLNB*, *GHSR*, *HDAC6*, *HMGA2*, *IGF1R*, *IHH*, *KDM6A*, *MAF*, *MATN3*, *NP R2*, *OBSL1*, *ORC1*, *PCNT*, *PDE3A*, *PDE4D*, *PIK3R1*, *POC1A*, *PTPN11*, *BLM*, *SHOX*, *SHOX2*, *SLC26A2*, *SMAD4*, *SRCAP*, *STAT3*, *TBX15*, *TRAIP*, *TRIM37*, *TRPS1*, *UBR1*, *XRCC4*, *COL 9A1*, *COL9A2*, *LTBP3*, *COL1A1*, *COL1A2*, *BMP1*, *CREB3L1*, *CRTAP*, *FKBP10*, *P3H1*, *PPIB*, *SERPINF1*, *SERPINH1*, *SP7*, *SPARC*, *TMEM38B*, *WNT1*, and *IFITM5*.

The BaseSpace pipeline (Illumina, San Diego, CA, USA, https://basespace.illumina.com/, accessed on 15 September 2020) and the TGex software LifeMap Sciences, Walnut, CA, USA, http://tgex.genecards.org/, accessed on 21 September 2020) were used for the variant calling and annotating variants, respectively.

Sequencing data were aligned to the hg19 human reference genome. Based on the guidelines of the American College of Medical Genetics and Genomics, a minimum depth coverage of 30X was considered suitable for analysis.

Mutations identified as pathogenic were confirmed by Sanger sequencing, following a standard protocol (BigDye Terminator v3.1 Cycle Sequencing Kit, Applied Biosystems by Life Technologies, Waltham, MA, USA).

## 3. Results

The exam showed a de novo missense heterozygous variant NM_000088.3: c.3235G>A, p.(Gly1079Ser) of the *COL1A1* gene.

This variant was not present in DNA samples of the parents. It was not reported in the frequency database of the general population and was described in the literature in association with OI [18,19] and could be classified as a pathogenic variant (class 5), according to the American College of Medical Genetics and Genomics (ACMG) guidelines [20].

## 4. Discussion and Review of Literature

We described a case of a young girl with a de novo missense heterozygous mutation c.3235G>A p.Gly1079Ser of *COL1A1* gene, already described in the literature in association with type I and IV OI [18,19], and a clinical phenotype compatible with a OI/EDS syndrome.

To the best of our knowledge, this is the first case of OI/EDS overlap syndrome associated with this heterozygous *COL1A1* variant.

According to Cabral et al. [21] and Malfait et al. [16], OI/EDS syndrome could be defined as a generalized connective tissue disorder characterized by features of both osteogenesis imperfecta (bone fragility, long bone fractures, blue sclerae, short stature) and Ehlers–Danlos syndrome (joint hyperextensibility, soft and hyperextensible skin, abnormal wound healing, easy bruising, vascular fragility).

Literature data about this complex and rare syndrome are very few. Most recent reports, made by Morlino et al. [13], described clinical and molecular features of 21 individuals, firstly ascertained by a clinical suspicion of EDS, in whom a wide spectrum of *COL1A1* and *COL1A2* heterozygous variants (that usually cause OI) were detected.

OI/EDS syndrome is clinically heterogeneous and no conventional criteria are still available to define this condition. The identification of these patients is consequently usually difficult: they are often classified within the variability of a broader definition of OI or diagnosed with a form of EDS featuring some of the minor clinical manifestations of OI [13].

The name of the syndrome has also been questioned and some authors proposed the broad nomenclature of “COL1-related overlap disorder” (C1ROD) to indicate an intermediate/mixed phenotype at the between of EDS and OI. This entity is characterized by the presence of causative variants in *COL1A1* and *COL1A2* genes in individuals with EDS typical phenotype (associated or not with signs of mild-moderate OI) that does not strictly coincide with a certain type of mutation of type I collagen gene [13].

The identification of causative variants in *COL1A1* or *COL1A2* genes is necessary for confirming the diagnosis of C1ROD.

Morlino et al. [13] identified a number of major and minor diagnostic criteria for C1ROD whose presence should be addressed to perform specific molecular testing.

Major criteria are represented by the presence of blue sclerae, flatfeet with valgus deformity, generalized joint hypermobility according to age, and significantly soft and doughy and/or hyperextensible skin.

Dolichostenomelia (arm span/height ratio > 1.05 in adult only); hearing loss; short stature (<2 DS); two or more atrophic (non-papyraceous) scars; two or more fractures in prepubertal age; two or more joint dislocations; and two or more injuries and/or ruptures of ligaments, tendons, and/or muscle were reported by Morlino et al. as minor criteria.

Molecular testing in index patients with suspected C1ROD should be performed in presence of three or four major criteria or two major criteria plus two or more minor criteria or one major criterion and five or more minor criteria [12].

We hypothesize that our case could be considered as an unusual variant of OI/EDS syndrome because of the coexistence of typical elements of EDS arthrochalase (such as joint hypermobility and mildly hyperelastic skin) in addition to the aforementioned OI phenotype.

In our case, we identified the presence of three major criteria (blue sclerae, generalized joint hypermobility according to age, and hyperextensible skin) and one minor criterion (short stature), so we decided to include molecular testing for C1ROD with NGS technology in our diagnostic workup.

NGS technology is considered a powerful approach to detect *COL1A1* and *COL1A2* mutations associated with connective tissue disorders, such as OI or EDS. Prior to its application, biochemical collagen analysis of fibrillar collagen proteins was used to identify the defect responsible for the phenotypical features observed in patients with suspected OI or EDS. Testing sensitivity is high for both types of investigations, but it is not clear if the sensitivity is additive [7].

To date, a combined approach with biochemical collagen analysis and NGS technologies could be considered in patients in whom the diagnosis in unclear, including those with OI/EDS overlap syndrome [11].

The majority of OI/EDS overlap syndrome patients displayed a heterozygous variant in the N-terminal part of the type I collagen helical region in either the α-1 or α-2 chain which affects—to some extent—proper processing of the N-propeptide [16,22], and results in the formation of small and weak collagen fibrils with irregular contours [20]. Glycine substitutions are frequently found in patients with a mixed OI/EDS phenotype [16] and condition clinical presentation.

This glycine residue appears to be evolutionally highly conserved if this change induces a small physicochemical difference between glycine and serine. Glycine residues within the Gly-Xaa-Yaa repeats of the triple helix domain are required for the structure and stability of fibrillar collagens [23,24,25]. Experimental studies show that this missense change (p.Gly1079Ser) results in minor helix destabilization [26] at the C-terminal region. For these reasons, this variant is classified as pathogenic.

Furthermore, as previously described [18], changes affecting glycine residues could be responsible of an OI phenotype characterized by low bone mineral density, short stature, dentinogenesis imperfecta, and scoliosis, as with our patient. On the other hand, EDS features seem to be more evident in individuals with abnormal/delayed type I procollagen N-propeptide processing, due to variants affecting the N-terminal of α-1 and α-2 procollagen [16].

In our case, the initial suspicion was about a mild form of OI. Additionally, the type of mutation found—classified as pathogenic variant (class V)—has been already reported to segregate with type I OI in several families [26,27,28,29]. First description was made in 1992 and interested Italian patients from the same family [26]. It has also been detected in many individuals and families with OI types I, III, and IV [19,30,31,32,33,34].

To date, 23 individuals with this p.Gly1079Ser mutation of *COL1A1* gene have been described in literature [4,6,19,28,30,32] and all identified as OI subtypes: 8 patients were diagnosed with type IV OI, 14 were classified as suffering from type I OI, and 1 case remained undetermined. There are both sporadic and familial cases and a broad phenotypic variability has been observed, also among the same family members. This different genotype–phenotype correlation is not clear and could be related to the influence of other genes and environmental factors on clinical presentation [35].

Short stature was a prominent feature of our patient, however her stimulated GH and basal IGF1 levels were normal and other endocrinological causes of short stature were ruled out because of the normality of hormone dosages.

Severe short stature may be common, but it is a not well-studied feature in OI patients which usually presents growth failure and compromised final height [36].

Growth impairment in OI patients usually starts since toddler age and is mainly due to the absence of osteoanabolic stimuli, secondary to defects in collagen structure that alter chondrocyte maturation and influence the so called “functional muscle-bone-unit” activity [37,38,39].

The severity of growth failure is strictly dependent on the OI severity, so it is more severe in patient with type 3 OI, who reach a final height of less than 100 cm [40].

This complex etiology could also explain the lack of effectiveness of attempts with GH treatment [41], whose secretion rate is usually normal [41] and whose action on growth velocity could depend probably on the underlying disease [42].

Short stature is rarely observed among EDS patients, excepting for the rare “Ehlers–Danlos syndrome, spondylodysplastic form type 3”, secondary to recessive loss-of-function variants in *SLC39A13*, a zinc transporter gene [43,44]; however, it has been described as a common clinical feature among patients with “OI/EDS overlap syndrome” (Table 1).

In our patient’s history, feeding problems had been a remarkable feature. In a cohort of patients with OI/EDS overlap syndrome [13] a condition of neonatal hypotonia was reported, but no mention of feeding problems was present. This element is increasingly found among patients with several genetic diseases, and often its presence could be a signal of a specific disease (e.g., Prader–Willi syndrome) [45].

We could not say whether in our girl this condition was related to neonatal hypotonia or whether it was a direct consequence of her genetic defect.

Additionally, the incipient neuroblastoma could justify its presence.

The association with a past history of neuroblastoma is another important element in our Case Report. Malignancies are rarely described among OI patients [46] and an alleged biochemical resistance to cancer of these individuals was suggested [47].

Osteosarcoma is the most frequent malignancies among OI patients [48,49] and might be probably secondary to the strong irradiation they are exposed to because of the great number of diagnostic radiographs of fractured bones, as reported in previous studies. Moreover, sporadic cases of tumors derived from breast, ovarian, colon, and gastric epithelium were described [45,50,51,52,53,54,55]. Furthermore, malignant tumors are also infrequent in patients with EDS and no cases of neoplasms have been, until now, reported in patients with OI/EDS syndrome.

NB is an embryonic developmental malignancy derived from sympathetic neural crest cells that usually affects children less than 18 months of age. It presents different phenotypes, that range from spontaneously regressive forms to metastatic and fatal disease, especially children older than 18 months at the time of diagnosis [56,57].

Stage 4 NB affects patients less than 18 months of age and it is characterized by the presence of metastasis to bone with or without manifestations in other metastatic sites [58]. Children with stage 4 nonamplified *MYCN* disease show a better outcome even with reduced treatment [59].

Genetic predisposition to NB development has been associated with common and rare genetic germline mutations [57]. At a somatic level, NB tumors show relatively few recurrent alterations, with point mutations in ALK (8–10%) and in ATRX (1–2%) being the most frequent [57]. A possible explanation for the association between *COL1A1* mutation and NB could be given by recent genomic analysis suggesting *COL1A1* playing a role in NB development. Indeed, our whole exome and deep sequencing study found somatic pathogenic mutations occurring in collagen type family genes—including *COL1A1*—in a subset of high-risk NBs [60]. While a cis-regulatory network analysis of NB reported *COL1A1* as a relevant gene regulated by a super enhancer involved in the modulation of NB cell identity [61].

Moreover, *COL1A1* is indicated as a predisposition gene in pediatric cancers [62]. Finally, *COL1A1* seems to also play a role in initiation and progression of other cancers, such as colon cancer [63], gastric cancer [64], esophageal cancer [65], and ovarian cancer [66].

We are not able to explain how the presence of *COL1A1* mutation could influence NB onset or progression in our patient. The link between both the diseases currently remains only a hypothesis based on the observation of a single case and further studies are needed to confirm our assumption.

## 5. Conclusions

The presence of *COL1A1* c.3235G>A variant might be suggestive of a diagnosis of OI/EDS overlap syndrome. We also could hypothesize that the observed association with the previous history of neuroblastoma is not incidental, but it could be influenced by the presence of *COL1A1* mutation, whose role in some cancers behavior and progression has been already described.

## Figures and Tables

**Figure 1 genes-13-00581-f001:**
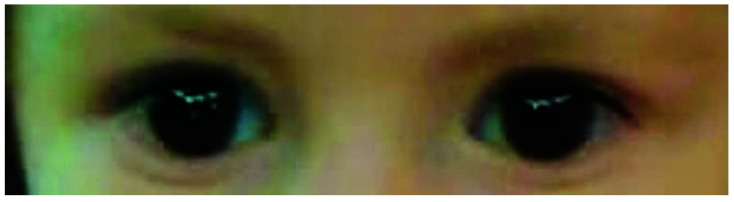
Clinical feature of the subject with OI/EDS overlap syndrome: Proband’s photographs showing blue sclerae.

**Table 1 genes-13-00581-t001:** Clinical features of patients with OI/EDS syndrome reported in literature.

References	No. of Case Described	Gene	Nucleotide Change	De Novo (DN)/Familial (F) Case	Blue Sclerae	Hyper Extensible Skin	Positive Beighton Score	History of Fractures	Altered Bone Density	Short Stature	Cardiac Valvular Defects	Easy Bruising	Hearing Loss	Joint Pain	Other Features
Morlino et al., 2020	21	*COL1A1* (6 cases)	c.2073delT	F	+	+	+	+	-	-	+	-	-	+	Piezogenic papules
			c.1243C>T	F	+	+	+	+	+	-	+	-	-	+	Flatfeet
			c.670G>A	F	+	+	+	+	-	+	NA	-	-	+	Muscle ruptures
			c.581G>C	F	+	+	+	-	-	-	NA	-	+	+	Microcornea progressive scoliosis
			c.326G>A	F	+	+	+	-	-	-	NA	-	-	+	Dystrophic scars
		*COL1A2* (15 cases)	c.577G>A	F	+	+/−	+	+	+	-	+	+	+	+	Chronic periodontitis, neonatal hypotonia
			c.432 + 5G>A		+	-	+	+	NA	-	-	-	-	-	---
			c.335G>T		+	-	+	+	-	-	NA	-	-	+	Flat feet, progressive scoliosis
			c.197G>A		+	+	+	+	-	-	NA	-	-	+	Dental crowding, high arched palate
			c.432+ + 4_432 + 7delAGTA	F	+	+	+	-	-	-	+	+	-	+	Flatfeet
			c133G>T	F	-	+	+	+	-	-	-	+	-	-	Myopia, gingival fragility
			c.316G>A	F	+	-	+	+	-	+	+	+	-	+	Myopia, high arched palate
			c.2755G>A	DN	+	-	+	+	-	+	NA	+	-	+	Flatfeet
Budsamongkol et al., 2019	1	*COL1A2*	c.3296G>A	DN	+	+	+	+	+	+	NA	-	NA	-	Brachydactyly, malocclusion dentinogenesis imperfecta, skeletal deformities
Lin et al., 2019	1	*COL1A1* *COL5A1*	c.2010delT c.5335A>G	F	+	-	+	+	NA	NA	NA	+	-	-	Prominent ears, atrophic scarring
Mackenroth et al., 2016	1	*COL1A1* *TNXB*	c.4006-1G>A c.7774G>A c.3637G>A	F	-	-	+	+	+	-	-	-	-	-	Severe muscular weakness, abnormally shaped vertebra
Malfait et al., 2013	7	*COL1A1*	c.563G>A	DN	+	+	-	+	+	+	-	+	NA	NA	Atrial septum defect, muscular hypotonia, arterial rupture
			c.607G>T	DN	+	+	+	+	+	+	-	+	NA	NA	Aortic dilation, inguinal hernia joint dislocation, kyphoscoliosis
		*COL1A2*	c.324+4delA	DN	+	+	+	+	+	+	+	+	NA	NA	Muscular hypotonia, joint dislocation
			c.587G>T	F	+	+	NA	-	+	NA	-	+	NA	NA	Muscular hypotonia, intracranial bleeding
			c.432 + 4_432 + 7delAGTA	DN	+	+	+	+	+	+	-	+	NA	NA	Joint dislocations
			c.587G>T	DN	+	+	+	+	+	+	-	+	NA	NA	Muscular hypotonia
			c.693+5G>A	DN	+	+	NA	+	+	-	NA	+	NA	NA	Pes planus, mild bowing of tibia and fibula
Present report	1	*COL1A1*	c.3235G>A	DN	+	+	+	-	+	+	-	-	-	-	Changes in tooth enamel, history of neuroblastoma, scoliosis

DN: de novo; F: familial; NA: not available.

## Data Availability

Not applicable.

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
