# Peer review of "Osteogenesis Imperfecta/Ehlers–Danlos Overlap Syndrome and Neuroblastoma—Case Report and Review of Literature"

_genes, 2022, doi:10.3390/genes13040581_

Round 1

Reviewer 1 Report

I think  that the authors have excellently addressed the problem related to genotype-phenotype correlation in EDS ad OI and have clearly dealt with an open problemn related to overlapping OI/EDS. overall, the manuscript may be to liking of readers.

Author Response

Dear Reviewer 1,

we would thank you for your appreciation and we are grateful to receive your positive comments. We performed an extensive  English language editing, as suggested. 

Reviewer 2 Report

Please explain more the interest of the case. The testing or is lacking, such as bone biopsies and definitions of OI is lacking. In addition The writing is a bit basic I would expand more the knowledge in regards to the overlap of these two diseases

Author Response

Point 1: Please explain more the interest of the case. 

Response

Dear Reviewer 2,

your comments are very interesting and we would thank you for your careful revision. We enriched the introduction section, also with OI definition (page 2 -3, lines 44-81) and modified the discussion according to your comments.

To the best of our knowledge, our case is the first description of OI/EDS syndrome associated with  the heterozygous COL1A1 variant c.3235G>A p.(Gly1079Ser), as reported in the manuscript (page 7, lines 220-221). 

Also the association with neuroblastoma is another important element in our Case Report, as cited in page 9 (lines 337-339). No cases of neoplasms have been, until now, reported in patients with OI/EDS syndrome (page 7, lines 344-
346).

Point 2:  The testing or is lacking, such as bone biopsies and definitions of OI is lacking. In addition The writing is a bit basic I would expand more the knowledge in regards to the overlap of these two diseases

In our case, OI/EDS overlap syndrome diagnosis was confirmed by  molecular analysis of COL1A1 gene, that revealed the presence of specific mutation in a patient with a clinical phenotype characterized by features of both OI and EDS. Prior to collagen genes screened by NGS technologies, biochemical collagen analysis of fibrillar collagen proteins was the starting point to identify the defect responsible for the phenotypical features observed in patients with suspected OI and EDS. Current indications suggest a combined approach with biochemical collagen analysis and NGS technologies only in patients in whom the diagnosis in unclear, including those with OI/EDS overlap syndrome (page 5, lines 228-236). In our patient the suspicion of OI/EDS overlap syndrome was very high, because the presence of three major criteria and one minor criteria for C1ROD proposed by Morlino et al.

Further information about OI/EDS overlap syndrome were added throughout the text.

As you suggested, we improved “Material and Methods” section by introducing a dedicated “Study design” and "Samples" paragraph (page 5, lines 122-125; page 6 ,lines 181-184). Results section was also expanded with a brief description of our patient’s variant (page 6, lines 210-214).